# A Rare Benzothiazole Glucoside as a Derivative of ‘Albedo Bluing’ Substance in Citrus Fruit and Its Antioxidant Activity

**DOI:** 10.3390/molecules29020302

**Published:** 2024-01-06

**Authors:** Chao Yang, Chuanxiu Yu, Qiang Li, Liangzhi Peng, Changpin Chun, Xiaolong Tang, Song Liu, Chengbo Hu, Lili Ling

**Affiliations:** 1National Citrus Engineering and Technology Research Center, Citrus Research Institute, Southwest University, Chongqing 400712, China; 17793610724@163.com (C.Y.); 15680831070@163.com (C.Y.); pengliangzhi@cric.cn (L.P.); chuncp@cric.cn (C.C.); 2Chongqing Key Laboratory of Environmental Materials & Remediation Technologies, Chongqing University of Arts and Sciences, Chongqing 402160, Chinabch1002@126.com (C.H.); 3National & Local Joint Engineering Research Center of Targeted and Innovative Therapeutics, College of Pharmacy, Chongqing University of Arts and Sciences, Chongqing 402160, China

**Keywords:** citrus, ‘albedo bluing’, purification, structural identification, antioxidant abilities

## Abstract

‘Albedo bluing’ of fruits occurs in many varieties of citrus, resulting in a significant reduction in their commercial value. We first presented a breakthrough method for successfully extracting and purifying the ‘albedo bluing’ substance (ABS) from citrus fruits, resulting in the attainment of highly purified ABS. Then, HPLC and UPLC-QTOF-MS were used to prove that ABS in the fruits of three citrus varieties (*Citrus reticulate* Blanco cv. ‘Gonggan’, ‘Orah’, and ‘Mashuiju’) are identical. However, the chemical structure of ABS remains elusive for many reasons. Fortunately, a more stable derivative of ABS (ABS-D) was successfully obtained. Through various analytical techniques such as HRESIMS, 1D and 2D NMR, and chemical shift calculation, ABS-D was identified as 2,4-dihydroxy-6-(*β*-D-glucopyranosyloxy)phenyl(5,6-dihydroxy-7-(*β*-D-glucopyranosyloxy)benzo[d]thiazol-2-yl)methanone, indicating that both ABS and its derivative belong to a rare category of benzothiazole glucosides. Furthermore, both ABS and ABS-D demonstrated potent antioxidant abilities. These findings lay the groundwork for further elucidating the chemical structure of ABS and the causative mechanism of the ‘albedo bluing’ phenomenon in citrus fruits.

## 1. Introduction

Citrus holds the distinction of being the most popular fruit globally. As of 2021, citrus production worldwide has reached 162 million tons [1]. Particularly in China, the citrus industry has grown remarkably since the beginning of the 21st century. In 2007, China surpassed Brazil for the first time in terms of citrus cultivation area and production, securing the top position globally [2]. Subsequently, in 2018, citrus production in China surpassed that of apples, solidifying the status as the fruit most widely produced in the country [3]. Despite the extraordinary progress in the Chinese citrus industry, an unsettling issue called ‘albedo bluing’ has emerged in significant citrus-producing regions such as Guangxi, Guangdong, Yunnan, and Sichuan. This phenomenon, known as ‘browning’, ‘brown-blue’, ‘moldy change’, and ‘inner mold’, initially presents as a light brown-blue discoloration in the albedo, near the fruit pith tissue. In milder instances, the discoloration is confined to the albedo tissue near the pith of the citrus fruit. However, in more severe cases, the brown-blue color spreads throughout the entire albedo, causing the normal light yellow or light white-yellow appearance to transform into a light brown-blue, dark brown-blue, or even black-blue hue (Figure 1A). The consequences of ‘albedo bluing’ are not merely limited to the fruit’s visual appearance; there are also adverse effects on the flavor, often resulting in noticeable off-odors. Consequently, the marketability of the affected fruits is significantly compromised. Growing concerns among consumers about the safety of consuming these fruits have turned ‘albedo bluing’ into a major issue in the citrus industry.

The ‘albedo bluing’ phenomenon was first documented in the United States in 1944. Since then, it has been observed in oranges, tangelos, and grapefruit across states such as Arizona, Florida, California, and Texas [4]. In recent years, China has also experienced instances of ‘albedo bluing’ in various citrus varieties, including *Citrus reticulate* Blanco cv. ‘Gonggan’, ‘Orah’, ‘Mashuiju’, ‘shatangju’, and ‘Murcott’. Researchers investigating this issue assert that the ‘albedo bluing’ in citrus fruits is not a result of pathogenic factors but rather a physiological disorder triggered by non-biological stress factors [4,5]. Initially, the ‘albedo bluing’ substance (ABS) was believed to be a natural water-soluble pigment resembling anthocyanins [4]. However, despite this assumption, no reports were available to confirm the structural identification of ABS in terms of anthocyanins. In more recent times, researchers have made new findings in this area. They observed that the extraction of blue albedo did not show the characteristic absorption peaks associated with anthocyanins, and no related substances were detected through LC-MS analysis [5].

The elusive nature of ABS, stemming from challenges in extracting and purifying it, has kept its chemical structure shrouded in uncertainty. Consequently, the causative factors and mechanisms of ‘albedo bluing’ in citrus fruits remain unclear, making it difficult to implement effective control measures during production. This study presents a breakthrough method for successfully extracting and purifying ABS from citrus fruits, resulting in the attainment of highly purified ABS and its derivative, ABS-D. The chemical structure of ABS-D has been successfully identified (Figure 1B), and this achievement serves as a strong starting point for further investigations aimed at determining the exact chemical structure of ABS and unraveling the underlying mechanisms responsible for the ‘albedo bluing’ phenomenon.

## 2. Results and Discussion

### 2.1. Extraction and Purification of ABS from ‘Gonggan’

In the context of citrus production, the variety known as ‘Gonggan’ exhibits the highest incidence and severity of ‘albedo bluing’. Due to this, ‘Gonggan’ was selected as the primary focus for studying the extraction and purification method of ABS. During the initial stages of the study, various organic solvents with different polarities and concentrations were tested for the extraction of ABS. However, it was found that the effective extraction of ABS could only be achieved under acidic conditions. Consequently, the optimized extraction method for ABS was established as follows: 1% hydrochloric acid in ethanol (*v*/*v*) as the extraction solvent, an extraction time of 2 h, and a liquid-to-solid ratio of 1:15 (*w*/*v*). The optimized extraction method was employed in the study to extract the blue albedo and normal albedo from ‘Gonggan’. The extraction solutions obtained from both samples were then analyzed using a spectrophotometer to identify the characteristic absorption wavelengths of ABS. As depicted in Figure 2A, only the extraction solution from the blue albedo displayed a significant absorption peak at 570 nm, indicating the presence of ABS in the blue albedo. A comparative analysis was performed using HPLC with 570 nm as the detection wavelength to further compare the presence of ABS and its content between the blue albedo and normal albedo. As shown in Figure 2B, in the HPLC analysis, a distinct absorption peak was observed at 17.09 min in the blue albedo extraction solution, representing the presence of ABS. On the other hand, a weak absorption peak appeared at 17.03 min for the normal albedo extraction solution, indicating a lower content of ABS in the normal albedo than in the blue albedo. The UV-Vis spectra of both absorption peaks observed in the HPLC analysis were consistent, confirming the successful extraction of ABS from the albedo of ‘Gonggan’. To address the issue of extraction solution fading during the rotary evaporation process, a method involving the addition of 5% sodium bicarbonate (*v*/*v*) to the extraction solution was employed to precipitate ABS (Appendix A). Following the precipitation step, the mixture was then subjected to filtration to obtain the crude extract of ABS.

Various chromatographic fillers were tested during further purification of the crude extract of ABS, including silica gel, macroporous resin, and polyamide. However, ABS showed a strong tendency to tail or could not be efficiently eluted, leading to significantly reduced recovery rates and purification efficiency. Finally, the HLB solid-phase extraction column was found to be highly effective in eliminating residual salts and certain hydrophilic impurities while also increasing the recovery rate of ABS (up to 95.2%) when using a minor concentration of hydrochloric acid in the eluent. Further purification of the ABS sample was carried out using Sephadex LH-20 column chromatography and semi-preparative HPLC. This purification process successfully yielded 3.7 mg of ABS with a purity ≥ 99% from 9958.00 g of ‘Gonggan’ albedo (Figure 2C). Notably, the color of the ABS solution was found to vary depending on the pH of the medium (Appendix A). At pH 1, the solution appeared purple; at pH 2–4, it appeared blue-purple; at pH 5–9, it appeared indigo blue; and at pH 10–12, it appeared cyan. These observations were distinct from the color changes seen in anthocyanins [6]. Additionally, it was noted that the pH of the ethanol extract from citrus albedo is around 6–6.5 [7], and under this pH the ABS solution primarily exhibited a blue color. This confirmed that purified ABS was responsible for the ‘albedo bluing’ phenomenon observed in citrus fruits. 

### 2.2. Comparative Analysis of ABS among Different Citrus Varieties

In addition to ‘Gonggan’, both ‘Mashuiju’ and ‘Orah’ are also known for having a higher incidence of ‘albedo bluing’. HPLC analysis was conducted to determine whether the ABS compounds in these three varieties are the same. As shown in Figure 3A,B, both ‘Mashuiju’ and ‘Orah’ displayed the characteristic ABS DAD absorption peak at approximately 16.91 and 16.92 min, respectively, similar to ‘Gonggan’. Furthermore, UPLC-QTOF-MS analysis (Figure 3C–E) revealed the presence of ABS molecular ion peaks at *m*/*z* 642.1135, 642.1134, and 642.1134, respectively, in ‘Gonggan’, ‘Mashuiju’ and ‘Orah’. These molecular ion peaks were detected at approximately 2.4 min in the chromatograms. The presence of identical ABS molecular ion peaks in all three varieties confirms that the ABS compound in each is the same chemical substance. 

### 2.3. Preparation of ABS-D

The ^1^H NMR analysis of ABS revealed the presence of conformational isomerism (Appendix A). Various deuterated solvents were tested to overcome this issue, but unfortunately none provided a satisfactory solution to address conformational isomerism. Moreover, ABS demonstrated chemical instability and was prone to fading, which made its analysis and characterization more challenging. Additionally, an extremely low abundance of fruit (approximately 1.5 × 10^−9^) posed significant hurdles for the large-scale purification of ABS. This limitation hindered the feasibility of conducting single-crystal induction experiments and precluded the use of X-ray for structural analysis, making the direct determination of ABS’s structure difficult.

An interesting observation was made during the initial stages of the ABS separation and purification process. The blue ABS solution gradually changed to a pale-yellow color when exposed to acidic conditions, and the conversion rate was affected by temperature and pH. To further investigate this phenomenon, the blue ABS solution was rapidly transformed into a yellow solution using 1% hydrochloric acid as the solvent and applying water bath heating. As depicted in Figure 4, during the heating process with durations ranging from 30 and 120 min, a new substance (ABS-D) emerged at approximately 10.8 min in the HPLC chromatogram. As the heating time was extended, the peak area corresponding to ABS-D gradually increased, while the peak area of ABS decreased proportionally. This indicated a transformation of ABS into ABS-D during the heating process. At a heating time of 120 min, ABS was completely converted into ABS-D, and no new absorption peaks were observed in the chromatogram, suggesting that ABS-D remained unchanged under the conditions of 1% hydrochloric acid and water bath heating. ABS-D was more stable than ABS, as it did not undergo further changes under these conditions. To maximize the purity of ABS-D, semi-preparative HPLC was utilized for its purification. This purification process yielded 2.9 mg of ABS-D with a purity ≥ 99%.

### 2.4. Structural Identification of ABS-D

The compound ABS-D, obtained as a pale-yellow powder, exhibited a noticeable blue shift in its UV-Vis spectrum compared to ABS. This shift indicated a significant disruption of the large conjugated system within the structure of ABS-D (Appendix A). HRESIMS analysis of ABS-D revealed a quasi-molecular ion peak at *m*/*z* 660.1213 [M+H]^+^ (C_26_H_30_NO_17_S, calc. 660.1229) (Appendix A), providing valuable information about the molecular formula of the compound, which was determined as C_26_H_29_NO_17_S with a total of 13 degrees of unsaturation. In the MS^2^ spectrum, a consecutive neutral loss of 162 Da (C_6_H_10_O_5_) was observed for ABS-D, generating characteristic ion fragments [M+H-C_6_H_10_O_5_]^+^ at *m*/*z* 498.0696 and [M+H-2C_6_H_10_O_5_]^+^ at *m*/*z* 336.0186 (Appendix A). These findings indicated that ABS-D is a di-*O*-hexoside, implying the presence of two hexose sugar units in its structure.

The ^1^H NMR spectrum of ABS-D recorded at 600 MHz in DMSO-*d*_6_ exhibited two characteristic anomeric proton signals at *δ*_H_ 4.80 (1H, d, *J* = 7.8 Hz) and 4.75 (1H, d, *J* = 7.8 Hz), indicating the presence of two sugar moieties in the compound. Additionally, twelve proton signals were observed in the range of *δ*_H_ 2.90–3.70, which were clearer in the range of *δ*_H_ 2.70–3.90 in CD_3_OD:CF_3_COOD = 20:1 (Appendix A), corresponding to the sugar units. In the ^13^C NMR spectrum recorded at 150 MHz in DMSO-*d*_6_, two characteristic sugar anomeric carbon signals were observed at *δ*_C_ 105.4 and 100.5, further confirming the presence of a diglycoside moiety in ABS-D. Ten oxygenated carbon signals were also observed in the range of *δ*_C_ 60–80, corresponding to the carbon atoms in the sugar moieties (Table 1). Based on these NMR data and a comparison with the relevant literature [8,9,10,11], the two glycoside moieties in ABS-D were considered *β*-D-pyranose glucose units.

Excluding the sugar proton signals, the ^1^H NMR spectrum, when combined with the HSQC spectrum, revealed four distinct proton signals at *δ*_H_ 9.79 (1H, s), 9.77 (1H, br s), 9.71 (1H, br s), and 9.38 (1H, s). Additionally, three aromatic proton signals were observed at *δ*_H_ 7.25 (1H, s), 6.12 (1H, br s), and 6.05 (1H, br s). The proton signals of aromatic moieties exhibit relatively low intensity, which is disproportionate to the proton signals of sugar. This observation may be attributed to the higher reactivity and susceptibility of aromatic protons to undergo deuterium exchange, particularly in methanol-*d*_4_ solvent where the deuteration rate is higher compared to DMSO-*d*_6_ (Appendix A). This phenomenon is commonly observed in polyhydroxy diaryl ketoside compounds [12,13,14]. Excluding the carbon signals of the two glucose moieties, the ^13^C NMR spectrum exhibited 14 carbon signals, which included one carbonyl carbon at *δ*_C_ 187.6 and 13 aromatic carbon signals. Notably, the HMBC correlations provided valuable insights into the structure of ABS-D. Specifically, HMBC correlations from the aromatic proton at *δ*_H_ 6.05 (1H, br s) to *δ*_C_ 160.5, 157.3, 108.0, and 94.0, and from the aromatic proton at *δ*_H_ 6.12 (1H, br s) to *δ*_C_ 160.5, 157.1, 108.0, and 96.5, indicated the presence of an A-ring structure in ABS-D (Figure 5). Additionally, HMBC correlations from the active proton at *δ*_H_ 9.79 (1H, s) to *δ*_C_ 160.5, 96.5, and 94.0, and from the active proton at *δ*_H_ 9.71 (1H, s) to *δ*_C_ 157.3, 108.0, and 96.5 further confirmed the A-ring structure. Furthermore, HMBC correlations from aromatic proton *δ*_H_ 7.25 (1H, s) to *δ*_C_ 147.9, 146.1, 139.7, and 122.7, from the active proton *δ*_H_ 9.77 (1H, s) to *δ*_C_ 147.9, 139.7, and 106.1, and from *δ*_H_ 9.38 (1H, s) to *δ*_C_ 147.9, 139.7, and 137.7 revealed the C-ring structure of ABS-D. The HMBC correlation from the sugar anomeric proton at *δ*_H_ 4.75 (1H, d, *J* = 7.8 Hz) to *δ*_C_ 157.1 and the correlation from the sugar anomeric proton at *δ*_H_ 4.80 (1H, d, J = 7.8 Hz) to *δ*_C_ 137.7 indicates that two glucose moieties are attached to the ABS-D structure at positions C-6 and C-3′, respectively. The carbonyl position was confirmed by the weak HMBC correlations from the aromatic protons at *δ*_H_ 6.12 (1H, br s) and *δ*_H_ 6.05 (1H, br s) to *δ*_C_ 187.6. Taking into consideration the molecular formula obtained from HRESIMS, the degree of unsaturation, and the NMR data reported for benzo[d]thiazole-2-yl(phenyl)methanone and benzo[d]isothiazol-3-yl(phenyl)methanone compounds found in the literature [15,16], three structural arrangements of the B-ring of ABS-D are proposed (Figure 6A). 

The application of NMR chemical shift calculations for the structural revision or assignment of intricate natural products is widely recognized. The combination of the traditional correlation coefficient (R^2^) method with the presently favored DP4+ analysis is the most frequently employed practice [17,18]. To determine the structure of the B-ring of ABS-D, NMR calculations were conducted for the three possible isomers—a, b, and c—using the mPW1PW91/6-311+G(d,p) level of theory. The linear correlation coefficient (R^2^) between the experimental and calculated ^13^C NMR data was analyzed, and the results indicated that isomer **a** was the most likely structure (Figure 6B). Furthermore, DP4+ statistical analysis was performed using the experimental and calculated ^13^C NMR data. The DP4+ probability analysis revealed that isomer **a** had a remarkably high probability of 99.99% (Figure 6C), confirming that isomer **a** was the most probable and accurate structure of ABS-D. 

### 2.5. Structure Prediction of ABS

A positive HRESIMS analysis of ABS provided an *m*/*z* value of 642.1133 ([C_26_H_28_NO_16_S]^+^, calc. 642.1129) (Appendix A), suggesting that the molecular formula of the compound is either C_26_H_28_NO_16_S or C_26_H_27_NO_16_S. The molecular mass of ABS is lower than that of ABS-D by 18.0080 Da or 17.0002 Da, indicating that ABS has one hydroxyl group or water molecule less than ABS-D. This suggests a hydration reaction may occur in ABS under acidic and heating conditions. The main structure of ABS is expected to be consistent with ABS-D, a benzothiazole glucoside compound. Possible structures of ABS are proposed, and they are all theoretically susceptible to transformation into ABS-D under acidic conditions (Figure 7). Benzothiazole compounds are relatively rare in nature, and their structural complexity ranges from benzothiazole itself and its simple derivatives to well-known molecules like firefly luciferin, and even more complex compounds such as thiazol-luciferin and dioxetane-quinone amine families [19]. However, compounds featuring the benzo[d]thiazole-2-yl(phenyl)methanone structure, like ABS-D, have not been found in nature. Nonetheless, benzo[d]thiazole-2-yl(phenyl)methanone and its derivatives have been successfully synthesized artificially and identified as potential therapeutic agents for various diseases, including estrogen-dependent disorders, hyperlipidemia, and certain neurological conditions, all while exhibiting strong metabolic stability [15,20,21]. 

### 2.6. Antioxidant Activities of ABS and ABS-D

The antioxidant capacities of the ABS and ABS-D compounds were assessed using DPPH, ABTS^+^, and FRAP assays (Table 2). In the DPPH assay, ABS (IC_50_ = 10.28 ± 0.15 μM) exhibited a much stronger efficacy than ascorbic acid (Vc) (IC_50_ = 34.64 ± 0.21 μM), while ABS-D (IC_50_ = 28.30 ± 0.34 μM) displayed a slightly greater antioxidant capacity than Vc. In the ABTS^+^ assay, ABS-D (IC_50_ = 19.63 ± 0.09 μM) showed significant antioxidant activity compared with Vc (IC_50_ = 46.26 ± 0.20 μM), while ABS (IC_50_ = 38.88 ± 0.39 μM) displayed potent antioxidant capacity with IC_50_ values close to that of Vc. Meanwhile, in the FRAP assay, ABS (314.61 ± 4.88 FeSO_4_·7H_2_O equivalent) and ABS-D (218.36 ± 1.96 FeSO_4_·7H_2_O equivalent) exhibited much higher ferric-reducing power ability compared to Vc (91.18 ± 2.50 FeSO_4_·7H_2_O equivalent). In conclusion, both ABS and ABS-D demonstrated good performance in in vitro antioxidant assays. The accumulation of secondary metabolites, such as phenolic compounds, terpenoids, and nitrogen-containing compounds, represents one of the mechanisms employed by plants in response to abiotic stress [22]. These compounds typically possess antioxidative properties and are capable of scavenging free radicals, thereby reducing the lipid peroxidation of cell membranes and enhancing the resistance of plants to abiotic stress [23,24]. It has been reported that the occurrence of ‘albedo bluing’ in citrus fruit may be associated with abiotic stress factors such as water stress, nutrient stress, and high temperatures, resulting in a weakening of tree vigor [4,5]. Furthermore, as tree vigor is restored, the phenomenon of ‘albedo bluing’ tends to diminish. Considering the potent antioxidant capability of ABS, we postulate that ABS might function as a secondary metabolite in response to abiotic stress in citrus fruits.

## 3. Materials and Methods

### 3.1. Materials and Reagents

In mid-January 2022, *Citrus reticulate* Blanco cv. ‘Gonggan’ fruits were harvested from Huangkeng Town, Renhua County, Shaoguang City, Guangdong Province, China (coordinates 25.06° N, 113.84° E). Similarly, *Citrus reticulate* Blanco cv. ‘Orah’ and ‘Mashuiju’ fruits were harvested in mid-February and mid-April 2020, respectively, from Wuming District, Nanning City, Guangxi Province, China (coordinates 23.17° N, 108.28° E). Immediately after harvesting, the fruits were transported to the Citrus Research Institute of Southwest University in Beibei District, Chongqing City, China, for further analysis. The sampling process involved segmenting the fruits into four sections, creating a cross shape, and separating the pulp from the peel. The albedo tissue of the peel was carefully scraped using a knife and collected in vacuum bags. These vacuum-sealed samples were stored at −20 °C in a freezer for subsequent extraction and analysis.

Ethanol, hydrochloric acid, sodium bicarbonate, and ammonium hydroxide were of analytical grade and obtained from Chuandong Chemical Co., Ltd. (Chongqing, China). Acetonitrile, methanol, formic acid, and phosphoric acid were of chromatographic grade and received from Kolon Chemical Co., Ltd. (Chengdu, China). DMSO-*d*_6_ and CD_3_OD were obtained from Cambridge Isotope Laboratories (Andover, MA, USA). Deionized water was obtained using a Milli-Q IQ7000 water purification system (Millipore, Milford, MA, USA). 1,1-diphenyl-2-picrylhydrazyl (DPPH), 2,2′-azino-bis (3-ethylbenzothiazoline-6-sulfonic acid) (ABTS), potassium persulphate (K_2_S_2_O_8_), and 1,3,5-tri(2-pyridyl)-2,4, 6-triazine (TPTZ) were purchased from Sigma-Aldrich (St. Louis, MO, USA).

### 3.2. Extraction and Purification of ABS

The following steps were considered to extract ABS from the fresh blue albedo sample of ‘Gonggan’. Freezing and grinding: A fresh blue albedo sample weighing 9958.00 g was frozen using liquid nitrogen and then ground into a powder using a grinder. Extraction: The powdered sample was subjected to two extractions using 1% hydrochloric acid in ethanol (*v*/*v*) at a ratio of 1:15 (*w*/*v*). The mixture was stirred magnetically at room temperature for 2 h during each extraction and the resulting extracts from both extractions were combined. pH adjustment: The pH of the combined extract was adjusted to 7 using 5% sodium bicarbonate (*v*/*v*). Precipitation: As a result of pH adjustment, the crude extract of ABS precipitated as a blue flocculent material.

The extraction of ABS from the crude extract was completed using an HLB solid-phase extraction column (500 mg/6 mL, Waters, Milford, MA, USA) following the activation and equilibration procedure described by Li et al. [25]. The crude extract of ABS was dissolved in 20% formic acid (*v*/*v*) and then filtered through a 0.45 μm PES membrane (Beckman Biologicals, Changde, China) to remove any particulate matter or impurities. The filtered sample solution was loaded onto the HLB column and pre-equilibrated. The elution process sequentially passed through 6 mL of water, followed by 35% methanol (*v*/*v*), to remove impurities from the column. Then, 6 mL of 0.04% hydrochloric acid in methanol (*v*/*v*) was used to elute ABS, resulting in the collection of the sample solution containing ABS. The sample solution obtained after elution was subjected to further separation using a pre-equilibrated MCX column. After the initial elution from the HLB column using 6 mL of 1% hydrochloric acid in methanol (*v*/*v*) to remove impurities, ABS was further eluted from the column using 12 mL of 5% ammonia in methanol (*v*/*v*). A Sephadex LH-20 column (40 × 600 mm, Cytiva, Sweden) was utilized to separate the ABS sample solution. The elution was performed using 30% methanol (*v*/*v*) at a flow rate of 0.5 mL/min. The blue band containing ABS was collected at regular intervals of 10 min using a BSZ-100 fraction collector (Qingpu Luxi Instrument, Shanghai, China). The collected fractions were then concentrated using a rotary evaporator (EYELA, Tokyo, Japan) and a freeze-dryer (Boyi Kang Laboratory Instruments Co., Ltd., Beijing, China).

The purified sample, which included ABS, underwent further purification using NP7000 semi-preparative HPLC (Hanbang Technology, Jiangsu, China) equipped with a C18 column (10 × 250 mm, 5 μm, Waters, Milford, MA, USA). The purification process involved using a linear gradient elution system with two solvents: deionized water (A) and acetonitrile (B). The gradient elution was as follows: 0–14 min, 10–22.5% B; 14–17 min, 22.5–100% B; 17–18 min, 10–100% B; 18–23 min, 10% B. The flow rate was set at 4 mL/min, and the detection of the sample was carried out at both 210 nm and 570 nm. As a result of this purification process, a total of 3.7 mg ABS was obtained, with its retention time (*t*_R_) measured as 16.97 min in the HPLC chromatogram.

### 3.3. HPLC Analysis of ABS

Gradient ABS in the samples was analyzed using a 1260 Infinity HPLC system (Agilent, Palo Alto, CA, USA) with a DAD detector, employing a Waters XBridgeTM C18 column (4.6 mm × 250 mm, 5 μm, Waters, Milford, MA, USA). The elution process involved a linear gradient using two solvents: 0.05% phosphoric acid (*v*/*v*) (A) and acetonitrile (B). The gradient elution was performed as follows: 0–25 min, 5–25% B; 25–30 min, 25–100% B; 30–31 min, 10–100% B; 31–38 min, 10% B. The column temperature was maintained at 35 °C during the analysis. For each injection, a sample volume of 5 μL was used, and the detection of ABS was carried out at two wavelengths: 240 nm and 570 nm. The flow rate was set at 1 mL/min during the analysis.

### 3.4. UPLC-QTOF-MS/MS Analysis of ABS

ABS molecular weight analysis was conducted using an ACQUITY-UPLC I-Class-G2-XS-QTOF system (Waters, Milford, MA, USA). The separation was performed on an ACQUITY UPLC BEH C18 column (2.1 mm × 100 mm, 1.7 μm, Waters, Milford, MA, USA). The elution was performed using a linear gradient with two solvents: 0.1% formic acid (*v*/*v*) (A) and acetonitrile (B). The gradient elution process was as follows: 0–8 min, 10–30% B; 8–9 min, 30–100% B; 9–10 min, 10–100% B; 10–12 min, 10% B. The flow rate during the analysis was set at 0.4 mL/min, and the column temperature was maintained at 35 °C. During the molecular weight analysis of ABS, a sample injection volume of 1 μL was used. Electrospray ionization (ESI) in positive-ion mode was employed for detection. The specific parameters for the ESI-MS analysis were as follows: capillary voltage, 3 kV; cone voltage, 40 V; collision voltage, 20 V for low-energy mode and 40 V for high-energy mode; desolvation temperature, 250 °C; ion source temperature, 100 °C; desolvation gas flow rate, 600 L/h; cone gas flow rate, 50 L/h.

### 3.5. Comparative Analysis of ABS among Different Citrus Varieties

In the comparative analysis of ABS from different citrus varieties (‘Gonggan’, ‘Orah’, and ‘Mashuiju’), 20.00 g samples of blue albedo were, respectively, weighed from the ‘Orah’ and ‘Mashuiju’ samples. For each variety, extraction, precipitation, and HLB solid-phase extraction pre-treatment of the samples were carried out following the procedures described in Section 3.2. After obtaining the purified samples of ABS from each variety, a comparative analysis was performed using two analytical techniques, HPLC and UPLC-QTOF-MS/MS, following the protocols outlined in Section 3.3 and Section 3.4, respectively.

### 3.6. Preparation of ‘Albedo Bluing’ Substance Derivative (ABS-D)

An amount of 3.0 mg of ABS was dissolved in 1 mL of 1% hydrochloric acid (*v*/*v*) and then placed in a water bath at 45 °C. The sample was subjected to different heating durations: 0, 30, 60, 90, and 120 min. During the heating process, the changes in ABS were monitored and analyzed using HPLC, following the method described in Section 3.3. The heating was stopped when the complete transformation of ABS into ABS-D occurred. After the transformation, ABS-D was separated using semi-preparative HPLC, following the conditions specified in Section 3.2. The resulting purified ABS-D product was obtained through rotary evaporation and freeze-drying techniques, producing 2.9 mg of pure ABS-D.

### 3.7. UV-Vis and NMR Analysis of ABS-D

To obtain the target compound’s UV-Vis spectrum, measurements were taken in the wavelength range of 210 to 800 nm using a UV-Vis spectrophotometer (Puxi General Instrument Co., Ltd., Beijing, China). Methanol was used as the solvent for these measurements. For the 1D and 2D NMR spectra, an AVANCE III 600 MHz NMR spectrometer (Bruker, Ettlingen, Germany) was utilized. The NMR spectrometer was equipped with a CPTCI cryogenic probe. The solvent signals DMSO-*d*_6_ (*δ*_H_ 2.50/*δ*_C_ 39.5) and CD_3_OD (*δ*_H_ 3.30/*δ*_C_ 49.0) were employed as internal standards during the NMR measurements.

### 3.8. Calculation of the Chemical Shifts of ABS-D

All density functional theory (DFT) calculations were carried out using the GAUSSIAN 16 series of programs. The density functional B3-LYP and a standard 6-31G(d) basis set were utilized for geometry optimizations. During these optimizations, DMSO was used as the solvent [26,27,28]. To evaluate solvation energies, a self-consistent reaction field (SCRF) with the PCM model was employed [29]. Furthermore, harmonic frequency calculations were performed for all stationary points to confirm them as local minima or transition structures. These calculations were crucial to derive thermochemical corrections for the enthalpies and free energies. All optimized structures were confirmed to be minima as they had zero imaginary frequency, indicating stability. The mPW1PW91 density functional was employed for the NMR calculations, along with a larger basis set of 6-311+G(d, p). DMSO was used as a solvent during these calculations. To determine the chemical shift (δ) values of ABS-D, Equation (1) was utilized.
(1)$${\delta =\sigma iso}_{\left(C-ABS-D\right)-}{\sigma iso}_{\left(C-TMS\right)}$$

In this equation, σiso_(C-ABS-D)_ represents the calculated magnetic screening value of the carbon atom in ABS-D, while σiso_(C-TMS)_ represents the calculated magnetic screening value of the carbon atom in tetramethylsilane (TMS).

### 3.9. Antioxidant Activity Assays

#### 3.9.1. DPPH Radical-Scavenging Activity

The DPPH assay was assessed using a previously described method with slight modifications [30]. Briefly, 50 μL of sample solutions at various concentrations (12.5–200 μg/mL) were mixed with 950 μL of DPPH solution (50 μg/mL) and recorded as A_s_. The control consisted of 50 μL of the sample (12.5–200 μg/mL) and 950 μL ethanol and was measured as A_c_, while the blank included 50 μL of ethanol and 950 μL of DPPH solution and was recorded as A_b_. All the mixtures were kept at room temperature for 30 min in the dark, and then the absorbance values were recorded later at 517 nm. Ascorbic acid (Vc) was used as a positive control, and the IC_50_ value represents the 50% inhibition ratio. The DPPH radical-scavenging capacity (C) was calculated by the following formula: $C=\frac{A_{b}-(A_{s}-A_{c})}{A_{b}}\times 100\%$.

#### 3.9.2. ABTS Radical-Scavenging Activity

The ABTS assay was performed following a previously described method with minor modifications [31]. In brief, the ABTS radical cation was generated by combining a 7 mM ABTS solution with 2.45 mM K_2_S_2_O_8_ in a 1:1 volume ratio. The resulting mixture was stored in the dark at room temperature for 24 h and then diluted with ethanol to achieve an absorbance of 0.7 ± 0.02 at 734 nm. Then, 100 μL of sample solutions at various concentrations (12.5–200 μg/mL) was mixed with 900 μL of ABTS solution and recorded as A_s_. The blank included 100 μL of ethanol and 900 μL of ABTS solution and was designated as A_b_. All mixtures were vortexed and kept at room temperature for 6 min, and then the absorbance values were recorded at 734 nm. Vc was used as a positive control, and the IC_50_ value represents the 50% inhibition ratio. The ABTS scavenging effect (E) was calculated using the following formula: $E=\frac{A_{b}-A_{s}}{A_{b}}\times 100\%$.

#### 3.9.3. Ferric-Reducing Antioxidant Power (FRAP) Assay

The FRAP assay was conducted following a previously reported method with minor modifications [31]. The FRAP solution was maintained in a water bath at 37 °C for 30 min. Then, 25 μL of sample solution (100 μg/mL) was mixed with 975 μL of the FRAP solution and incubated in the dark at 37 °C for 10 min. Subsequently, absorbance values were measured at 593 nm. A standard curve (y = 0.6008x − 0.0069, R^2^ = 0.999) was generated using ferrous sulfate (FeSO_4_·7H_2_O). The antioxidant capacity of the samples was expressed as FeSO_4_·7H_2_O-equivalent, with Vc serving as the positive control.

## 4. Conclusions

In this study, we successfully established a method for extracting and purifying ABS from citrus fruits. Through this method, we obtained highly pure ABS from the blue albedo of ‘Gonggan’, confirming its presence in ‘Orah’ and ‘Mashuiju’ as well. However, determining the exact structure of ABS remains challenging. To address this issue, we subjected ABS to acidic and heated conditions, forming a more stable yellow compound (ABS-D). Further extraction and purification of ABS-D resulted in a highly pure product. Its chemical structure was identified as 2,4-dihydroxy-6-(*β*-D-glucopyranosyloxy)phenyl(5,6-dihydroxy-7-(*β*-D-glucopyranosyloxy)benzo[d]thiazol-2-yl)methanone. It is believed that ABS also belonged to the class of benzothiazole glucosides. Furthermore, both ABS and ABS-D exhibited potent antioxidant abilities in in vitro antioxidant assays, implying that ABS might be a substance within citrus fruits that responds to environmental stress. This study lays a strong foundation for future research aimed at fully elucidating the chemical structure of ABS and understanding the causative mechanism of the ‘albedo bluing’ phenomenon in citrus fruits.

## Figures and Tables

**Figure 1 molecules-29-00302-f001:**
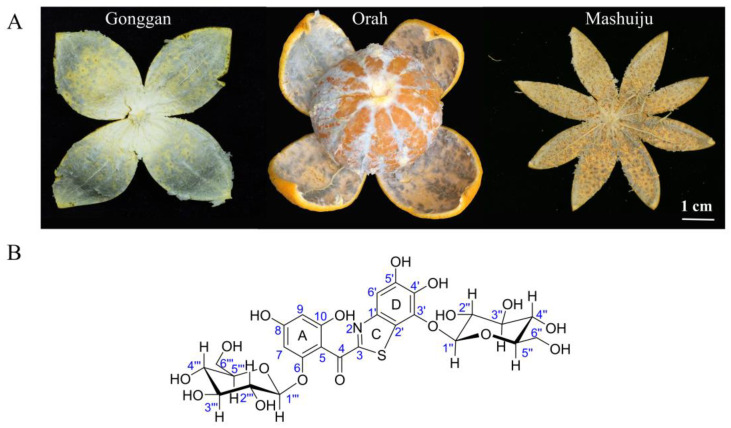
(**A**) Illustrations of the typical ‘albedo bluing’ phenomenon in ‘Gonggan’, ‘Orah’, and ‘Mashuiju’. (**B**) Chemical structure of ABS-D.

**Figure 2 molecules-29-00302-f002:**
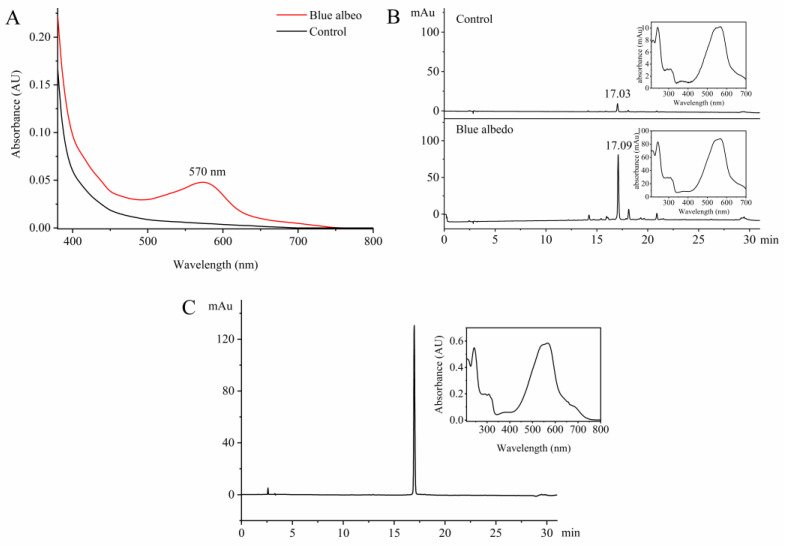
Analysis of UV-Vis and HPLC during the isolation and purification of ABS from ‘Gonggan’. (**A**) UV-Vis spectrum of the extraction solution; (**B**) HPLC analysis of the extraction solution (monitored at 570 nm); (**C**) HPLC analysis (monitored at 240 nm) and UV-Vis spectrum of the purified ABS.

**Figure 3 molecules-29-00302-f003:**
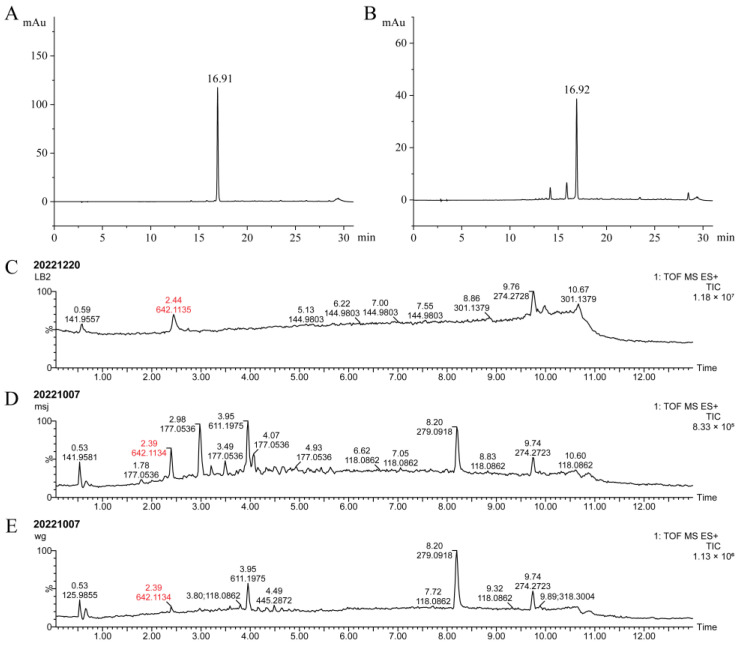
Comparative analysis of ABS in ‘Gonggan’, ‘Mashuiju’ and ‘Orah’. (**A**) HPLC analysis (monitored at 570 nm) of the albedo extract of ‘Mashuiju’; (**B**) HPLC analysis (monitored at 570 nm) of the albedo extract of ‘Orah’; (**C**) UPLC-QTOF-MS analysis of the pure ABS extracted from ‘Gonggan’; (**D**) UPLC-QTOF-MS analysis of the pre-processed extract of ‘Mashuiju’; (**E**) UPLC-QTOF-MS analysis of the pre-processed extract of ‘Orah’.

**Figure 4 molecules-29-00302-f004:**
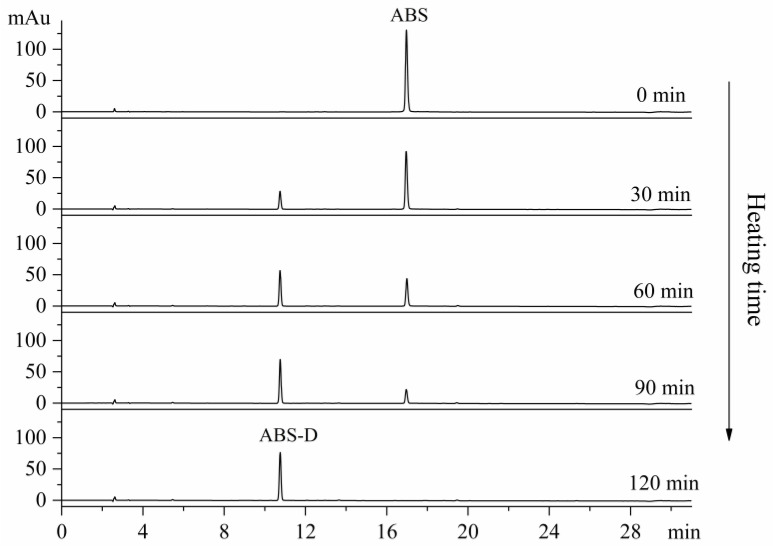
HPLC analysis of ABS converted to ABS-D (monitored at 240 nm).

**Figure 5 molecules-29-00302-f005:**
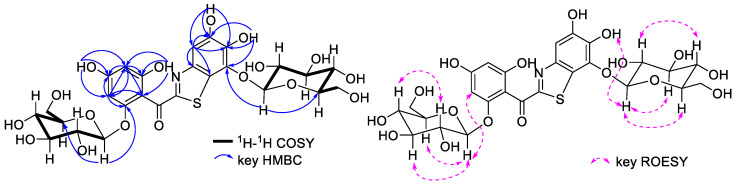
COSY, key HMBC, and ROESY correlations of ABS-D.

**Figure 6 molecules-29-00302-f006:**
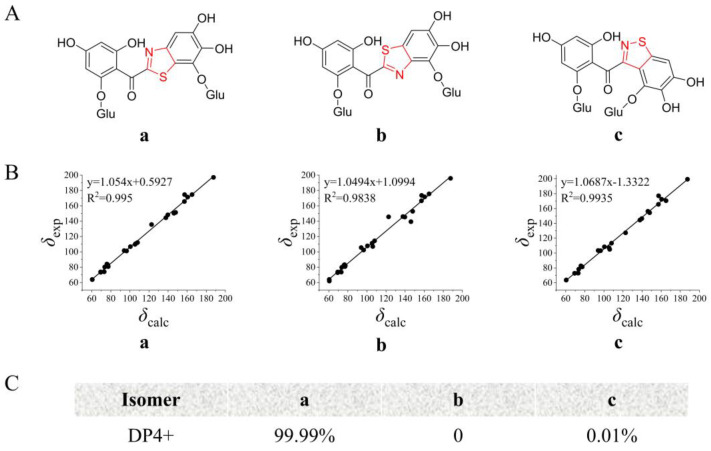
^13^C NMR chemical shift calculation for three possible isomers of ABS-D. (**A**) Structure of three possible isomers of ABS-D; (**B**) linear correlation between calculated and experimental values of three possible isomers of ABS-D; (**C**) DP4+ probability analysis for three possible isomers of ABS-D.

**Figure 7 molecules-29-00302-f007:**
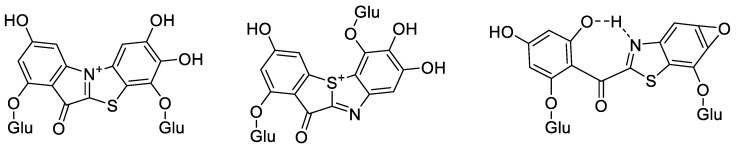
Possible chemical structures of the ABS.

**Table 1 molecules-29-00302-t001:** ^1^H NMR (150 MHz) and ^13^C NMR (600 MHz) spectra of ABS-D in DMSO-*d*_6_.

Position	*δ*_H_ (*J* in Hz) ^a^	*δ* _C_	^1^H-^1^H COSY ^b^	HMBC	ROESY
2					
3		165.1			
4		187.6			
5		108.0			
6		157.1			
7	6.12, br s	94.0		5, 6, 8, 9	1‴, 8-OH
8		160.5			
8-OH	9.79, s			7, 8, 9	7, 9
9	6.05, br s	96.5		5, 7, 8, 10	8-OH, 10-OH
10		157.3			
10-OH	9.71, br s			5, 9, 10	9
1′		146.1			
2′		122.7			
3′		137.7			
4′		139.7			
4′-OH	9.38, s			3′, 4′, 5′	5′-OH, 1″
5′		147.9			
5′-OH	9.77, s			4′, 5′, 6′	4′-OH, 6′
6′	7.25, s	106.1		1′, 2′, 4′, 5′	5′-OH
1″	4.80, d (7.8)	105.4	2″	3′, 3″, 5″	3″, 5″, 4′-OH
2″	3.35, t (7.9)	73.6	1″, 3″	1″, 3″	4″
3″	3.27, t (7.6)	75.9	2″, 4″	2″, 4″	1″
4″	3.26	69.3	3″, 5″	3″, 6″	2″, 6″
5″	3.24	77.3	4″, 6″	1″	1″
6″	3.68, br d (10.7)	60.5	5″	4″, 5″	4″
	3.55, dd (11.6, 4.3)			4″, 5″	4″
1‴	4.75, d (7.8)	100.5	2‴	6, 3‴, 5‴	7, 3‴, 5‴
2‴	2.93, t (8.2)	73.1	1‴, 3‴	1‴, 3‴	4‴
3‴	3.17, t (8.9)	76.5	2‴, 4‴	2‴, 4‴	1‴
4‴	3.06, t (9.2)	69.4	3‴, 5‴	3‴, 6‴	2‴, 6‴
5‴	3.22	77.0	4‴, 6‴	1‴	1‴
6‴	3.64, br d (10.4) 3.43, dd (11.7, 5.3)	60.6	5‴	4‴, 5‴ 4‴, 5‴	4‴ 4‴

^a^ NMR signals, which may arise from overlapping or complex multiplicity, are reported without specifying the exact multiplicity. ^b^ Determined in CD_3_OD:CF_3_COOD = 20:1.

**Table 2 molecules-29-00302-t002:** Antioxidant activity of ABS, ABS-D, and ascorbic acid.

Compounds	DPPH (IC_50_ μM) ^a^	ABTS^+^ (IC_50_ μM) ^a^	FRAP ^a,b^
ABS	10.28 ± 0.15	38.88 ± 0.39	314.61 ± 4.88
ABS-D	28.30 ± 0.34	19.63 ± 0.09	218.36 ± 1.96
Ascorbic acid ^c^	34.64 ± 0.21	46.26 ± 0.20	91.18 ± 2.50

^a^ Values represent means ± SD (*n* = 3). ^b^ The antioxidant capacity of the samples is expressed as Fe_2_SO_4_·7H_2_O-equivalent. ^c^ Positive control.

## Data Availability

Data are contained within the article and Appendix A.

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
