# Peer review of "A Rare Benzothiazole Glucoside as a Derivative of ‘Albedo Bluing’ Substance in Citrus Fruit and Its Antioxidant Activity"

_molecules, 2024, doi:10.3390/molecules29020302_

Round 1
Reviewer 1 Report (Previous Reviewer 2)
Comments and Suggestions for Authors
The authors have made all the required changes and corrections. I recommend accepting this manuscript in present form.
Author Response
Comments 1: The authors have made all the required changes and corrections. I recommend accepting this manuscript in present form.
Response 1: Dear reviewer, Thank you for providing valuable suggestions on our manuscript, and we appreciate your recognition of our research efforts.
Reviewer 2 Report (New Reviewer)
Comments and Suggestions for Authors
1. Page 6, Line 183. According to the 1H NMR spectrum (Fig. S8), there are 15 proton signals in this range.
Author Response
Comments 1: Page 6, Line 183. According to the 1H NMR spectrum (Fig. S8), there are 15 proton signals in this range.
Response 1: Dear reviewer, each hexose sugar bears 7 hydrogens (one is anomeric, and the rest is for other carbons). According to the integrations from δH 2.90-3.70 ppm, there are 15 hydrogens in 1H NMR with DMSO-d6 solvent (δH 2.50 ppm), of which 12 hydrogens belong to the hexose moieties. Three remaining hydrogens belong to the moisture of DMSO-d6 (δH 3.30 ppm).
Reviewer 3 Report (New Reviewer)
Comments and Suggestions for Authors
In this study, Ling and coworkers investigated the chemical properties of the widespread 'albedo bluing' substance in citrus fruits, a factor contributing significantly to their diminished commercial value. The authors proposed a novel method for successfully extracting and purifying the elusive 'albedo bluing' substance (ABS) from citrus fruits, resulting in a highly purified form. Through HPLC and UPLC-QTOF-MS analyses, they established the identical nature of ABS in the blue albedo extraction of three citrus varieties. The distinctive high-intensity HPLC peak for ABS in blue albedo extraction, compared to a very low-intensity peak for normal albedo extraction, indicates ABS as a potential marker for 'albedo bluing.' The study utilized bicarbonate precipitation, filtration, HLB solid-phase extraction, and semi-preparative Sephadex LH-20 HPLC column chromatography, proving effective in obtaining a highly pure form of ABS. The pH-dependent indigo-blue colour of ABS in a solution with pH 5-9, along with its blue colour within the pH range of the ethanol extract from citrus albedo (6-6.5), confirms its role in the 'albedo bluing' phenomenon in citrus fruits. Despite successful ABS extraction, its elusive chemical structure determination is attributed to factors like conformational isomerism, low abundance in natural sources, and chemical instability. However, the intriguing transformation of ABS into a more stable derivative (ABS-D) under acidic conditions and heating is characterized as a benzothiazole glucoside derivative. ABS-D is identified as 2,4-dihydroxy-6-(β-D-glucopyranosyloxy)phenyl(5,6-dihydroxy-7-(β-D-glucopyranosyloxy)benzo[d]thiazol-2-yl)methanone through various spectroscopic techniques. Both ABS and ABS-D exhibit potent antioxidant abilities, providing more reason for further exploration of ABS's chemical structure and understanding the causative mechanism of the 'albedo bluing' phenomenon in citrus fruits.
Overall, the manuscript presents a comprehensive investigation into the extraction, purification, and structure determination of the elusive 'albedo bluing' substance (ABS) in citrus fruits. By analyzing mass spectrometric data and time/temperature-dependent HPLC data, particularly the transformation into the stable derivative ABS-D under dilute acid conditions, the authors conclude that ABS also belongs to the class of benzothiazole glucosides. As both ABS and ABS-D exhibit potent antioxidant abilities in vitro antioxidant assays, the study suggests that ABS might be a substance within citrus that responds to environmental stress. This work establishes a foundation for future research on ABS's chemical structure and the causative mechanisms behind the 'albedo bluing' phenomenon in citrus fruits, emphasizing the need to address this issue in the citrus industry. However, the manuscript's significance relies on the isolation of pure ABS and ABS-D and the correct prediction of their structures. As the authors acknowledge the inability to determine the structure of ABS due to conformational isomerism, determining the structure of ABS-D becomes crucial. The authors have recorded all the necessary spectra required for the structural and stereochemical elucidation of ABS-D. However, their lucid interpretation in the text and representation of the structure of the molecules in the spectrum in the supporting information with clear assignments will enhance the manuscript's comprehensibility. Therefore, while the manuscript is well-structured and contributes significantly to understanding 'albedo bluing' in citrus fruits, following improvements are suggested to enhance comprehensibility of the manuscript before it can be recommended for publication in Molecules.
1. Determined chemical structures of the compound should be given in the inset of each spectrum to enhance the readability of the manuscript.
2. In few 13C NMR spectra, all the carbons are not properly visible, and in some cases, peaks are not properly marked with exact chemical shift. Presenting a magnified portion of highly populated peaks in the inset can be very helpful for comprehending spectral data.
3. As C-H cosy, HMBC spectra have been obtained, specific assignment of the proton-attached carbon is possible. It will great if the structure of the compound is presented with proper assignment of the peaks to the corresponding carbons and protons.
4. The 13C NMR (150 MHz, CD3OD) spectrum of ABS-D (Figure S9) and the 13C NMR (150 MHz, DMSO-d6) spectrum of ABS-D (Figure S10) look drastically different in terms of chemical shifts, peak numbers, and variation in peak intensity. Authors should provide some justification for such drastic solvent effect.
5. In ABS-D, none of the aromatic carbons is chemically equivalent. But its 13C NMR (150 MHz, DMSO-d6) spectrum (Figure S10) showed some highly intense peaks at the aromatic region, compared to the proton-attached carbohydrate carbons, though most of these are quaternary carbons and expected to give smaller peaks. It will be good if authors give some suitable justification for such an unexpected high intensity.
Author Response
Comments 1: Determined chemical structures of the compound should be given in the inset of each spectrum to enhance the readability of the manuscript.
Response 1: Dear reviewer, we have included the chemical structure of ABS-D in each spectrum. Thank you for your suggestion.
Comments 2: In few 13C NMR spectra, all the carbons are not properly visible, and in some cases, peaks are not properly marked with exact chemical shift. Presenting a magnified portion of highly populated peaks in the inset can be very helpful for comprehending spectral data.
Response 2: Dear reviewer, we have enlarged the high-density peaks in 1H NMR and 13C NMR (Figure S7-10) of ABS-D as much as possible, hope you can accept it.
Comments 3: As C-H cosy, HMBC spectra have been obtained, specific assignment of the proton-attached carbon is possible. It will great if the structure of the compound is presented with proper assignment of the peaks to the corresponding carbons and protons.
Response 3: Dear reviewer, we have incorporated proton and carbon assignment information in the 1H-1H COSY (Figure S11) and HMBC (Figure S14) spectra. Thank you for your valuable suggestions.
Comments 4: The 13C NMR (150 MHz, CD3OD) spectrum of ABS-D (Figure S9) and the 13C NMR (150 MHz, DMSO-d6) spectrum of ABS-D (Figure S10) look drastically different in terms of chemical shifts, peak numbers, and variation in peak intensity. Authors should provide some justification for such drastic solvent effect.
Response 4: Dear reviewer, we speculate that the occurrence of this difference is attributed to the presence of tautomeric isomerism involving the hydroxyl groups and their vicinal positions in ABS-D. Methanol is a protonating solvent that exacerbates tautomeric isomerism, while DMSO effectively suppresses it. Additionally, during NMR testing with methanol as the solvent, we added a minimal amount of trifluoroacetic acid, which has a certain inhibitory effect on tautomeric isomerism.
Comments 5: In ABS-D, none of the aromatic carbons is chemically equivalent. But its 13C NMR (150 MHz, DMSO-d6) spectrum (Figure S10) showed some highly intense peaks at the aromatic region, compared to the proton-attached carbohydrate carbons, though most of these are quaternary carbons and expected to give smaller peaks. It will be good if authors give some suitable justification for such an unexpected high intensity.
Response 5: Dear reviewer, we speculate that the cause of this phenomenon aligns with that of question 4, where the suppression of tautomeric isomerization occurs when DMSO is used as the solvent. As a result, the peak intensity of certain quaternary carbons is stronger than expected.
Reviewer 4 Report (New Reviewer)
Comments and Suggestions for Authors
I include pdf versión with some mistakes that I found in the manuscript.
Why did you mention in the discussion Table 1 and Table 1S, which is the difference between them?
Could you obtain a 1H-15N HMBC to discard options in your propose for ABS-D?
After that you answered these question your manuscript could be published.

Author Response
Comments 1: I include pdf versión with some mistakes that I found in the manuscript.
Response 1: Dear reviewer, We appreciate your careful review of our manuscript, and we have addressed the points you highlighted. Thank you for your valuable feedback.
Comments 2: Why did you mention in the discussion Table 1 and Table 1S, which is the difference between them?
Response 2: Dear reviewer, Table 1 and Table S1 present the NMR data of ABS-D measured in DMSO-d6 and CD3OD: CF3COOD=20:1, respectively. We provide these two sets of NMR data simultaneously because the signals of the sugar protons in ABS-D are clearer in CD3OD: CF3COOD=20:1 than in DMSO.
Comments 3: Could you obtain a 1H-15N HMBC to discard options in your propose for ABS-D?
Response 3: Dear reviewer, we sincerely apologize for not having the conditions to perform the 1H-15N HMBC analysis for ABS-D. However, we believe that by comparing the intensities of the signals at 6'H in the 1H-13C HMBC with those at 2'C and 1'C, we can make a preliminary assessment of the connection positions of N and S. Additionally, with the integration of NMR calculations, we consider that this approach is suitable for determining the structure of ABS-D.
Round 2
Reviewer 3 Report (New Reviewer)
Comments and Suggestions for Authors
The author has implemented the recommended modifications and provided appropriate justifications for apparent anomalies in the 13C NMR. The manuscript is now deemed acceptable for publication in Molecules.
This manuscript is a resubmission of an earlier submission. The following is a list of the peer review reports and author responses from that submission.
Round 1
Reviewer 1 Report
Comments and Suggestions for Authors
In this work ‘albedo bluing’ (ABS) was obtained in highly pure from the blue albedo of ‘Gonggan’. This is responsible of light white-yellow appearance and adverse effects on the flavor in the pith of the citrus fruit.
The ABS and ABS-D compounds were extracted and purified by HPLC technique. The last compund was obtained under heating in acid conditions. The structure of both compound were strongly determined from NMR and MS techinques. Also their antioxidant activities were analyzed.
This manuscript has high scientific rigor and it was writed in correct form includin all experimental details. For this reason this manuscript qualify as acepting in the present form.
Reviewer 2 Report
Comments and Suggestions for Authors
This article by Li, Ling, and co-workers describes an interesting study on a recurrent problem in citric production around the world, Albedo bluing. The authors evaluated the presence of AB in three different citrus varieties, Citrus reticulate Blanco cv. Gongga, Orah, and Mashuiju, using different chromatographic techniques. This study established that AB was identical among the three citrus varieties. In addition, the authors could determine the chemical structure of a derivative of the compound involved in the AB effects, establishing that it is a rare benzothiazole glucoside. In addition, the authors determined the antioxidant behavior of ABS and its derivatives, demonstrating an important activity. This experiment allowed the authors to propose that ABS could be a consequence of abiotic stress in citrus.
This is a good article that provides important information in the field of agriculture, food production, and the isolation of natural products. From a technical point of view, the article is of good quality; therefore, I recommend that this article be accepted after MINOR REVISIONS.
Some considerations must be taken into account:
- ABS-D is mentioned in the manuscript, but its chemical structure does not appear until page 8. This is confusing for the reader. The authors should include a figure indicating the proposed structure for ABS-D in the initial pages of the article.
- The proposed mass for ABS-D is 660.1229, whereas for ABS, the authors propose a mass of 642.1133. As the authors mentioned in 2.5 Structure Prediction of ABS, this means that a hydration process occurs in the transformation of ABS to ABS-D. Revising the proposed structure of ABS-D, the question is… Where? It would be interesting for the readers to propose a possible position for this hydration or a possible mechanism for the hydration reaction of ABS.
- Have the authors revised the other possible biological activities of ABS and ABS-D? In this sense, antibiotic activity could be a good candidate.
- The ABS-D structures depicted in Figure 5 are horrible. Some atoms overlap, and the quality is not good. Special mention should be given to the arrows used in the HMBC experiments, which are confusing and impossible to follow. The authors have to improve it.
- Several typing mistakes were detected in the article. An extensive revision is required.
- Several typing mistakes were detected in the article. An extensive revision is required.
Reviewer 3 Report
Comments and Suggestions for Authors
This manuscript describes the extraction and purification process of ABS from citrus fruits, the characterization of ABS and ABS-D, and the antioxidative activities of the obtained ABS and ABS-D.
A few concerns for the authors.
1. Line 173, “An interestin The compound ABS-D obtained as a pale yellow powder”. It looks like “An Interestin” should not be there.
2. On page 6, figure 6 would not be necessary if the authors have good HMBC and HMQC data. 1’C and 2’C should have different chemical shifts because one of them connected to N and the other one connected to S. If the authors found that there was an HMBC correlation be the C connected to S with an aromatic H, then among the three proposed structures of ABS-D, only a is possible.
3. On page 10, session “2.5 Structure Prediction of ABS”, are there any possible structures for ABS?